# A Machine-Learning–Blockchain-Based Authentication Using Smart Contracts for an IoHT System

**DOI:** 10.3390/s22239074

**Published:** 2022-11-23

**Authors:** Rajkumar Gaur, Shiva Prakash, Sanjay Kumar, Kumar Abhishek, Mounira Msahli, Abdul Wahid

**Affiliations:** 1ITCA, Madan Mohan Malaviya University of Technology Gorakhpur, Gorakhpur 273016, India; 2ITD, Rajkiya Engineering College Azamgarh, Deogaon 276201, India; 3CSED, National Institute of Technology Patna, Patna 800005, India; 4Telecom Paris, Institut Polytechnique de Paris, 91120 Palaiseau, France

**Keywords:** IoHT, smart contract, ML-based, training set, blockchain, SVM, CP-ABE, secure system

## Abstract

Nowadays, finding genetic components and determining the likelihood that treatment would be helpful for patients are the key issues in the medical field. Medical data storage in a centralized system is complex. Data storage, on the other hand, has recently been distributed electronically in a cloud-based system, allowing access to the data at any time through a cloud server or blockchain-based ledger system. The blockchain is essential to managing safe and decentralized transactions in cryptography systems such as bitcoin and Ethereum. The blockchain stores information in different blocks, each of which has a set capacity. Data processing and storage are more effective and better for data management when blockchain and machine learning are integrated. Therefore, we have proposed a machine-learning–blockchain-based smart-contract system that improves security, reduces consumption, and can be trusted for real-time medical applications. The accuracy and computation performance of the IoHT system are safely improved by our system.

## 1. Introduction

Amazing developments in internet technology have made IoT indispensable to every facet of human life. Technologists and the common man are now excited and eager to use IoT to make their lives easy and luxurious. Recently, people have been keen to use IoT-based smart networks, including smart homes, smart offices, smart marketplaces, etc. This is because of the high availability of data, sharing nature and time utilization. Vulnerabilities such as data forgery, privacy, integrity and third-party involvement are the constraints which stop or limit the use of IoT-based smart networks. Blockchain-technology involvement has been seen as an ay of light to overcome or minimize the different issues of IoT. Conventional smart homes with a centralized structure are obviously associated with many security issues and vulnerabilities. A suitable combination of IoT and blockchain is the most attractive area for researchers. This paper presents a short overview which outlines the various shortcomings of IoT-based networks and advancements of IoT networks using blockchain technology. Specifically, we consider the issues related to smart homes and their security at different layers. A possible architecture is proposed with certain advancements. This architecture is likely to improve data integrity at the gateway layer of smart homes [1,2].

Blockchain is developing to be a secure and trustworthy medium for secure data communication in the economic sector, supply-chain management, nutrition, energy, the IoT, and healthcare. This paper reviews existing surveys and applications available for the medical system using blockchain and machine-learning technology. Similarly, this work proposes numerous workflows in the medical ecosystem using machine-learning and blockchain techniques for more suitable security and data management. The machine-learning-and-blockchain-based applications secure medical data, manage patients and doctors, access data and control systems, verify research data, and perform exchanges for financial auditing and transparency.

IoT devices are an asset to home automation, involving heating, lighting, air conditions and security systems such as cameras. The long-term benefits of IoT are energy savings by automatically ensuring lights and electronic devices are switched off or by making the homeowner in the home aware of the use of devices by sending a few informative signal alerts [3].

Security is the most serious and major concern about adopting Internet-of-things (IoT) technology, because expeditious development is occurring without any relevant consideration of the contingent and profound security and privacy challenges involved and the legislative changes which are mandatory [4].

In fact, blocks are the fundamental concepts in blockchain technology. Blocks are small sets of transactions which have already taken place inside the system. In a chain, a new block saves the records of the prior transaction using the SHA-256 hash of the previous transaction. Hence, it connects blocks and makes a chain, as the name blockchain implies. Computationally, it is difficult to create the blocks, and they take multiple specialised processors and a valid amount of time to generate. However, the creation of a block is difficult and to deface one block, a previous block must be tampered with, which then has to experience the same process to alter it completely; therefore, blockchain technology is tamper-proof [5].


**Our contribution**


We propose a machine-learning–blockchain-based multidimensional architecture for IoHT which detects malware, classifies data, and provides lightweight security for healthcare monitoring systems in real-time.The machine-learning layer integrates with the proposed security techniques for blockchain-based smart contracts for transforming the data into the IoHT application.The feature selection of patients’ information is based on machine-learning techniques such as SVM for threat detection and the classification of the IoHT information.The proposed security system assesses and analyses the IoHT model’s performance in terms of authentication, identification, and privacy.

In our threat model, we consider three security objectives: authentication, access control and non-repudiation in the context of IoHT. In fact, several research papers have illustrated various IoT device-authentication mechanisms and techniques, to ensure the device identities and authentication of IoHT users.There are different types of attacks targeting IoHT systems presented in many studies [6]. In most cases, adversaries exploit security vulnerabilities in IoHT devices to perform sybil attacks, cyberattacks, and the use of malicious users and malicious devices. That is why, in this paper, we tackle the authentication security issue. Since we are dealing with very sensitive data such as patient information, X-ray images, etc., we consider the need to establish an access control layer to refine access to such data. We aim to add more confidence to the health data by adding a user-authorisation mechanism to avoid data disclosure. In this work, we also address the threat where a user is not able to deny certain shared data and executed activities. All the proposed security mechanisms could be used as proof in case of a future problem or traceability need.

The rest of the paper is organized as follows: Section 2 establishes the preliminaries for the SVM and Blockchain security protocols. In Section 3, an overview of the related work is presented. The proposed work is based on machine learning and blockchain-based security in Section 4. The experimental results and implementations are provided in Section 5. Finally, the conclusions of this paper are in Section 6, with a look toward our future work.

## 2. Preliminaries

Let two multiplicative cyclic groups be $G_{A}$ and $G_{B}$ with prime order p. In group $G_{A}$, g is a generator and $\hat{e}$ a bilinear map, such that: $\hat{e}:G_{A}\times G_{A}\to G_{B}$, and bilinear map has some properties such as:1.**Bi-linearity:**$\forall s,t\in G_{A}$ and $x,y\in Z_{p}$; then,$\hat{e}:\left(g^{x},g^{y}\right)$ = $\hat{e}{\left(s,t\right)}^{x,y}$2.**Non-degeneracy:**$\hat{e}\left(g,g\right)\neq 1$.Let $G_{A}$ be a bilinear group; if the group’s operation is in $G_{A}$, that is called symmetric group and the $\hat{e}$ is a bilinear map such that3.**Symmetric:**$\hat{e}:\left(g^{x},g^{y}\right)$ = $\hat{e}{\left(g,g\right)}^{x,y}$ = $\hat{e}:\left(g^{y},g^{x}\right)$

## 3. RelatedWork

A blockchain-based data-protection gateway was proposed by Lee et al. to guarantee the confidentiality, stability, and authenticity of data. Their suggested blockchain-based smart-home gateway includes layers such as the device layer, the machine layer, and the cloud layer. Appliances, devices, and sensors that collect and keep track of data from an intelligent home environment make up the device layer. The information coming from the device layer is stored in the machine layer’s second tier and is made available to the users as required. The cloud layer, which is the final layer, is responsible for recording the ID for each machine gateway and the data that each gateway operates. Blocks are fabricated so users can access their data anytime, whenever required. IoT devices in smart homes are assigned a unique ID using the SHA2 encryption algorithm to maintain the confidentiality and authenticity of the smart-home gateway [7,8].

Babou et al. found that current edge computing technology does not ensure the very low latency needed by real-time applications. They proposed a novel three-tier architecture with home edge devices whose main objective is to decrease latency and improve transmission quality by integrating micro-cells at the level of the personal server [9].

Xia Yang et al. proposed a hybrid-based blockchain authentication scheme for smart-home automation. They found that the central server cannot operate large-scale devices with high efficiency. They also showed that the system suffered from malicious devices and presented a physically secure authentication scheme [10].

Samrah et al. found that the traditional security scheme has failed to address the unique security concerns related to smart homes. They introduced a simple and secure smart-home framework on the basis of consortium blockchain [11].

Li Da Xu et al. examine the key problems of IoT such as security, privacy and reliability, which are limited in their development on a large scale, and they found that blockchain provides a decentralized and transparent database for security to a better level. Blockchain provides confidentiality, maintains the integrity of data by providing encryption and also minimizes various attacks and reduces transaction cost [12].

According to Manzoor et al., using a blockchain-based economizing for information sharing and a proxy-based re-encryption system for secure information transfer is preferable. The proposed methods use IoT sensors and additions from information storage. The system is free of outside interference [13].

Marko et al. suggest IoT appliances have inadequate computing complexity to maintain robust security with an encryption algorithm. The simple interface security gateway architecture and blockchain enable decentralization and authentication [14].

Many researchers, including Ana, Reyna, et al., M. A Naser et al., and Ali Dorri et al., have also conducted a broad survey on blockchain, its integration with IoT, challenges, and opportunities. They found that to ensure proper security in IoT, ethereum blockchain can be used. Cloud computing using a blockchain with distributed, secure, and confidential properties can deliver a reliable solution to smart homes with machines that capture environmental information. To minimize problems regarding data storage and its confidentiality in IoT smart homes, a three-tier architecture where a blockchain-based cloud service manages the data storage of an intelligent home is shown to provide greater security [15,16,17]. In both Bitcoin and Hyperledger Fabric, the suggested batch-verification approach can be successfully integrated. To lessen the difficulty of identifying erroneous signatures when the outcome of batch verification is false, we also integrate group testing approaches with our proposed batch-verification methodology. The incorporation of new tools into ECDSA to aid batch verification is an important area for future study [18].

### 3.1. Machine Learning

Machine learning is crucial to the growing field of data science. In data-mining initiatives, algorithms are trained using statistical techniques to produce classifications or predictions and uncover essential insights [19,20]. These sensitivities then affect how decisions are made by applications and businesses, hopefully having a significant impact on growth. As long as big data continues to flourish and grow, data scientists will need more. Data is required to choose the best business inquiries and the data required to answer them [1,21,22].

Input as a first layer, one or more secret layers as the second, and an output as a third layer; these systems’ three layers of nodes that make up neural networks are sometimes referred to as machine-learning approaches (ANNs). The threshold and weight of each node, or artificial neuron, move with it and are connected to other nodes. Assume any node is turned on and starts sending data to the layer below it in the network when its output hits a certain threshold. Otherwise, no data is transmitted from that node to the following tiers of the network.

This technology enables people to transact securely with one another via an approved secure and decentralized system without a negotiator. Machine learning can assist with several limitations of blockchain-based systems and their other capabilities. These machine-learning and blockchain-technology approaches can be combined to provide effective and worthwhile outcomes. In order to ensure security for IoMT applications, blockchain technology should be considered and how machine learning skills can be integrated with a process based on blockchain technology analyzed.

The blockchain uses machine-learning algorithms, which excel at learning, to build chains that are more hopeful than those created previously. The distributed ledger of the blockchain may be made more secure with the help of this integration. Additionally, the computational power of ML can be leveraged to speed up data-sharing routes and decrease the time required to determine the golden nonce. The decentralised-data-architecture feature of blockchain technology also allows us to create more, better machine-learning algorithms.

Machine-learning models can manipulate the information saved in the blockchain network to forecast the future or analyze medical data. The IoMT application, where information is collected from various sources, including detectors, intelligent gadgets, IoT-based appliances, and the blockchain application, functions as a crucial element. With an enormous amount of medical information, the ML-based model can be used for real-time information analytics or predictions and maintain the massive amounts of information in the blockchain network, which allows for lower errors in the networks. The data will not contain any missing values, replications, or noise, which is essential for the machine-learning model to deliver high accuracy.

### 3.2. Blockchain

Blockchain was created in 2008 by Satoshi Nakamoto and contains a network of autonomous networks that controls a time-stamped collection of damaging confirmation documents. A chain of blocks combined with basic cryptography makes up the blockchain architecture. The key ideas driving blockchain technologies’ operation are inflexibility, decentralization, and transparency [1]. The private blockchain network is centered on the IoMT network, depicted in Figure 1.

The medium consists of three layers: the first layer of data sources which relate to the network and emit user information, the second layer of a private blockchain network powered by BL–ML-based techniques which releases predictive analysis of thousands of files, and the third user layer where client node with data connect, from the IoMT approach, with a group of medical devices that impersonate the pieces of information [23,24].

### 3.3. A Comparative Study and Discussion

In this section, a comparative study is presented in Table 1 and Table 2. These tables present a brief account of the resolution of various issues in IoT and blockchain, and their possible solutions proposed by various researchers using blockchain technology, especially in reference to IoT-based smart homes. Table 2 is self-explanatory: the relevance of blockchain technology in tackling various issues associated with IoT. On the basis of the above comparison and a detailed examination of references cited therein, it may be safely concluded that the use of blockchain in smart homes can appreciably resolve and improve various issues such as the privacy, integrity, authenticity and security of data in IoT-based homes can be efficiently resolved to a greater extent [19]. From a comprehensive study of various papers, it has been observed that different researchers have proposed the use of blockchain technology in various ways, such as using consortium blockchain for security, a distributed cloud layer based on blockchain, using hybrid type of blockchain and a secure gateway architecture for providing authenticity [21,25].

Table 1 analyzes the various machine-learning-based schemes applied in blockchain-based applications. The machine-learning-based approach uses various classification methods and detection methods for the analysis of massive hospital data of patients. The IoHT system reduces the various processes of data processing, and this system also securely transmits the information using the CP-ABE scheme. This table analyses the various attributes of the schemes, such as accuracy, approach, types of data used, aims, classes of data sets, the platform used, and schemes.

Table 2 represents the comparison of various schemes used in blockchain-based security with different platforms. This table uses many factors when comparing various schemes, such as the problem identification, techniques and technology used in the paper.

## 4. Proposed Scheme

The section discuss the various techniques used in an IoMT system for secure data transformation.

### 4.1. ML-Based Data Classification

Let us suppose a classification issue in medical data using the different techniques with labeled training data—$\left\{\left(x_{i},y_{i}\right)\ldots \left(x_{m},y_{m}\right)\right\}$. The objective is to find a function ∮(x) such that $\oint \left(x_{i}\right)\approx y_{i}$$\forall_{i}\in \left\{1\ldots m\right\}$. To search ∮(x) in the space of all possible processes is computationally infeasible and prone to overfitting. Therefore, a procedure class F is specified, and the most suitable ∮(x) in F identified [34,35].

Therefore, defining a function class F and how to measure the closeness of $\oint \left(x_{i}\right)\;\mathrm{and}\;y_{i}$ provides an upgrade to various algorithms such as:

1.If the function class F is a linear combination of all data points, and the metric to approximate $\oint \left(x_{i}\right)\;\mathrm{and}\;y_{i}$ is hinge loss, then a support vector machine (SVM) is utilized.2.If the function class FF is a conditional possibility model with a specific independence system, and the metric maximizes the probability of data, then a Naive Bayes classifier should be used.

The paper used the support vector machine for medical data classification and data attributes used for the CP-ABE scheme for the security system in the blockchain.

#### Working of SVM

An approach for supervised binary classification is the support vector machine (SVM). SVM generates an (N-1) dimensional hyperplane to split points into two classes if any provided group of points of two classes is in an N-dimensional place. Let some attributes of two types in applications be linearly divisible. SVM will find a straight line splitting those attributes into two categories and be as nearby as conceivable to all those attributes. Although A is located as far away from all those traits as possible, lines A and B separate the two classes of attributes properly. SVM will choose the separating hyperplane. The margin is shown in this Figure by the pale blue region surrounding lines A and B; it is the space between the hyperplane and the nearest point multiplied by two. Another technique preserves the hyperplane in the midpoint of the margin. The perfect hyperplane will be produced by the highest margin.

A classification scheme typically uses training and testing data from some data models, even though users are not required to comprehend the SVM’s underlying theory. Each model in the training set has numerous/attributes as well as one/target value (class labels) (features). With only the attributes provided by the testing set, the goal of SVM is to create a model that predicts the target value of data instances.

### 4.2. Description

Let S be a training set given as S = $\left\{\left({\vec{x}}_{i},\;y_{i}\right)\right\},\;\left(i=1,2,\ldots ,l\right)$, where every ${\vec{x}}_{i}\;\in \;R^{n}$ is a data sample and each $y_{i}$ is its class. SVMs’ binary classification considers the existence of dividing surface (decision hyperplane) between two types labelled −1 and 1 i.e., here, $y_{i}=\left\{-1,1\right\}$. The decision hyperplane’s assumption and its coefficients’ determination by SVM is either linear-separable or non-linear-separable.

#### 4.2.1. Linear Classification

The linear hyperplane used to separate training data linearly can be given as $\vec{w}.\;\vec{x}+b=0$, where $\vec{w}\in R^{n}$ is an n-dimensional vector norm to represent a hyperplane direction and b∈R is hyperplane location. Both (+ive) and (−ive) observations individually lie on the complementary side of a matching corroborative hyperplane for the individual classification due to being linear detachable and can be defined as
(1)$$y_{i}\left(\vec{w}.{\vec{x}}_{i}+b\right)\geq 1,\;i=1,2,\ldots ,l$$

Distance from data sample $\vec{x_{i}}$ to the separator is $r=\frac{\vec{w}.\;\vec{x}+b}{∥w∥}$. **Note:**
∥…∥ specifies norm square. The separator’s maximum margin ρ is the distance between support vectors. The SV is those data selections that are closer to the verifying hyperplane. The classification of nonlinear attachable data via a linear hyperplane with support vector machine increases to the quadratic modification issue, i.e., to obtain $\vec{w}$ and b so that $\rho =\frac{2}{∥w∥}$ is increased ∀$y_{i}\left(\vec{w}.{\vec{x}}_{i}+b\right)\geq 1,\;i=1,2,\ldots ,l$. The classification also reformulates and finds $\vec{w}$ and *b*, so that $\Phi \left(w\right)={∥w∥}^{2}$ is minimized for all $y_{i}\left(\vec{w}.{\vec{x}}_{i}+b\right)\geq 1,\;i=1,2,\ldots ,l$. The justification for the above quadratic optimization issue produces a dual problem where a Lagrange multiplier $\alpha_{i}$ is associated with every inequality constraint.
(2)$$W\left(\vec{\alpha }\right)=\sum_{i=1}^{l}\;\alpha_{i}-\frac{1}{2}\;\sum_{i=1}^{l}\sum_{j=1}^{l}\;y_{i}\;y_{j}\;\alpha_{i}\;\alpha_{j}\;\left({\vec{x}}_{i}\;.\;{\vec{x}}_{j}\right)$$
is maximized such that
$$\sum_{i=1}^{l}y_{i}\alpha_{i}=0$$
and $\alpha_{i}\geq 0,\;\forall \;\alpha_{i},\;i=1,2,\ldots ,l$. Each non-zero $\alpha_{i}$ indicates that the corresponding ${\vec{x}}_{i}$ is a support vector. The $\vec{w}$ and b are the separating hyperplane coefficients and are computed as
(3)$$\vec{w}=\sum_{\vec{x_{i}}\in SV_{s}}\alpha_{i}\;y_{i}\;\vec{x_{i}}$$
(4)$$b=\frac{1}{N_{SV_{s}}}\sum_{\vec{x_{i}}\in SV_{s}}b_{i}$$
(5)$$b_{i}=y_{i}-\vec{w}.{\vec{x}}_{i}$$
where i=1,2,…,l, $SV_{S}$ implies support vectors and $N_{SV_{s}}$ is the number of support vectors.

#### 4.2.2. Soft Margin Classification

It is possible to identify training errors by tracking how noisy information selections differ from the relevant supportive hyperplane or the perfect linear separability circumstance. Slack coefficient $\xi_{i}$ can be added to allow these noisy samples to result in a soft margin. The modified formulation of the quadratic optimization problem involving slack coefficients can be given so as to find $\vec{w}$ and *b* such that $\Phi \left(w\right)={∥w∥}^{2}+C\sum_{i=1}^{l}\xi_{i},\;C>0$ is minimized for all $y_{i}\left(\vec{w}.{\vec{x}}_{i}+b\right)\geq 1-\xi_{i},\;\xi_{i}\geq 0\;\mathrm{and}\;\;i=1,2,\ldots ,l$. The solution to the above quadratic optimization problem can be given as:(6)$$W\left(\vec{\alpha }\right)=\sum_{i=1}^{l}\;\alpha_{i}-\frac{1}{2}\;\sum_{i=1}^{l}\sum_{j=1}^{l}\;y_{i}\;y_{j}\;\alpha_{i}\;\alpha_{j}\;\left({\vec{x}}_{i}\;.\;{\vec{x}}_{j}\right)$$
and is maximized such that
$$\sum_{i=1}^{l}y_{i}\alpha_{i}=0$$
and $0<\alpha_{i}\leq C,\;\forall \;\alpha_{i},\;i=1,2,\ldots ,l$, where *C* is the distance of error of data sample ${\vec{x}}_{i}$ from its correct place. The separating hyperplane coefficients b can be computed as
(7)$$b_{i}=y_{i}\left(1-\xi i\right)-\vec{w}.{\vec{x}}_{i}$$

Utilizing the decision-making function, it is possible to specify the type of each test model or the side of the decision limitation where a new individual data sample is encountered as $g\left(\vec{x}\right)=sign\left\{\vec{w}\;.\;\vec{x}\;+\;b\right\}$.

#### 4.2.3. Non-Linear Classification

A linear decision hyperplane can be generated afterward by non-linearly mapping the training data samples onto a high-dimensional feature space. The partitioning hyperplane will generate an exact type in the attribute space by minimizing training errors. A non linear map $\Phi \;:\;R^{n}\;\to \;H$ is performed. $\vec{w}$ is correspondingly mapping via ΦintoH. Thus, the concerned squared model maximized in the limit of the fork now becomes ${∥\Phi \left(w\right)∥}^{2}$ and the equation of the hyperplane can be presented as $\Phi \left(\vec{w}\right).\Phi \left(\vec{x_{i}}\right)+b=0$. The SVMs use a kernel, a positive (semi-)definite function, to find a solution to the optimization issue and can be given as $K\left(\vec{x},\vec{y}\right)=\Phi \left(\vec{x}\right).\Phi \left(\vec{y}\right)$, where $\;\;\vec{x},\;\vec{y}\;\in \;R^{n}$. The SVMs utilize a pair of standard kernels [36,37,38]:The p-degree Polynomial classifier: $K\left(\vec{x},\vec{y}\right)={\left(1+\vec{x}.\vec{y}\right)}^{p}$The radial basis function classifier: $K\left(\vec{x},\vec{y}\right)=\exp\left(-\frac{∥\vec{x}-\vec{y}{∥}^{2}}{2\sigma^{2}}\right)$ where p and σ are hyper parameters of SVMs. σ is the distance between the closest points with different classifications.

The solution to the nonlinear quadratic optimization problem can be given as:(8)$$W\left(\vec{\alpha }\right)=\sum_{i=1}^{l}\;\alpha_{i}-\frac{1}{2}\;\sum_{i=1}^{l}\sum_{j=1}^{l}\;y_{i}\;y_{j}\;\alpha_{i}\;\alpha_{j}\;K\left({\vec{x}}_{i}\;.\;{\vec{x}}_{j}\right)$$
and is maximized such that
$$\sum_{i=1}^{l}y_{i}\alpha_{i}=0$$
and $0<\alpha_{i}\leq C,\;\forall \;\alpha_{i},\;i=1,2,\ldots ,l$. The decision procedure is computable as,
(9)$$g\left(\vec{x}\right)=sign\left(\sum_{\vec{x_{i}}\;\in \;SV_{s}}\alpha_{i}y_{i}K\left(\vec{x_{i}}.\vec{x}\right)+b\right).$$

#### 4.2.4. Multi-Class Support Vector Machines

Multiple binary classifiers must be built in order for multi-class SVMs to solve machine tasks individually. Each sample in the test is assigned a class based on the decision functions of those binary classifiers as a whole. Three distinct systems, one-against-one, one-against-all and decision-directed acyclic graph, are available for this purpose.

**One against all:** With the one-against-all (1-aa) method, k-classifiers are created. Each $j^{th}$ SVM treats all training samples labeled with *j* as positive, whereas all other training samples are treated as hostile. The decision hyperplane quadratic optimization problem that best distinguishes the samples of result *j* from all other samples in the training set may be written as
(10)$${minimize\;∥w∥}^{2}+C\sum_{i=1}^{l}\xi_{i}^{j}$$
based on
$$y_{i}\left(K\left({\vec{w}}^{j},{\vec{x}}_{i}\right)+b\right)\geq 1-\xi_{i}^{j},\;\xi_{i}^{j}\geq 0$$
for j=1,2,…,kandi=1,2,…,l.A sample for a test’s class $\vec{x}$ is given by decision function $class\left(\vec{x}\right)=argmax_{j=1,2,\ldots ,k}$$\left(K\left({\vec{w}}^{j},\vec{x}\right)-b^{j}\right)$.**One against one:** The one-against-one (1-a1) method creates SVMs with a $\frac{k(k-1)}{2}$ size. Every $j^{th}$ machine learns from data from classes *j* and *m*, where samples labelled with class *j* are viewed as positive and samples labelled with class *m* are viewed as negative. The decision hyperplane quadratic optimization problem that best distinguishes scenarios with outcome *j* from the samples with outcome *m* in the training set may be written as
(11)$${minimize\;∥w∥}^{2}+C\sum_{i=1}^{l}\xi_{i}^{jm}$$
based on
$$y_{i}\left(K\left({\vec{w}}^{jm},{\vec{x}}_{i}\right)+b\right)\geq 1-\xi_{i}^{jm},\;\xi_{i}^{jm}\geq 0$$
for j,m=1,2,…,kandj≠mandi=1,2,…,l.Each new sample $\vec{x}$ calculated by applying the sign of the decision function for every SVM is classified using a voting technique. The vote for the $j^{th}$ class is increased by one if the sign indicates that the sample is in class J; otherwise, the ballot for the $m^{th}$ class is increased. Finally, the selection is from the group that received the most votes. If two types acquire the same number of ballots, the classes with the lower index are picked.**Decision-directed acyclic graph:** Training in the DDAG technique is identical to the 1a1 technique. The coefficients correlation $\frac{k(k-1)}{2}$ to SVMs are developed; the subsequent graph approach is utilized to specify the class for a test sampling $\vec{x}$. Each node of the graph has an index of connected classes and considers the list’s first and last elements. The list at the source node has all *k* classes, and when traversed from node to node, the elimination of one class from each corresponding list is perfomred until leaves are reached. The scheme starts at the root node. At each node, *i* vs. *j*, refer to SVM trained on data from classes *i* and *j*. The class of $\vec{x}$ is computed following the sign of the corresponding decision function applied to $\vec{x}$. If the sign says a sample is in class i, the node is exited via the right edge; otherwise, the left edge. This eliminates the wrong class from the list and proceeds via the corresponding edge to test the first and last classes of the new list and node. The class is given by the leaf in the route.

### 4.3. Integration Machine Learning in Blockchain-Based IoHT

This subsection discusses the ML-based secure blockchain to an IoHT system.

It is a distributed database spread throughout an Internet of Hospital Things network’s nodes.Since there is no single authority over it, it is a decentralized technology. A large number of nodes manage the IoHT network.It guarantees visibility and confidentiality.All transactions are approved and authenticated on the network since it implements a proof-of-work consensus mechanism.

The model needs data for machine learning to build models and make predictions based on patterns and searching. The ML basis produces data run in the blockchain system. This MLBC-based technique effectively and securely stores data. Figure 2 shows the data processing and functioning data.

### 4.4. Implementation of Real-Time Blockchain-Based Security System in IoHT

**Attribute-based encryption:** Attribute-based encryption (ABE) is used for the real-time transmission of encrypted data keys to users, then each data key requires encryption only once utilizing a unique (global) ABE public key, and each user satisfying the access policy can decrypt using their own ABE private decryption key [39]. ABE encryption essentially consists of hiding the data key with a mask, with it being retrieved by authorized decryption keys. The ABE-encrypted data-key size and communication complexity are linear in the number of attributes, not the users. Attributes and access policies are produced in combined ABE encryption and decryption functions, so that only the users with the right private decryption keys, which satisfy the access policies for given data, can decrypt, hence, obtain the data key and the data; inherent flexibility allows dynamic access policies and/or attributes [32]. The CP-ABE scheme uses the same security services as the scheme shown in Figure 3:Control: The user should be able to fully supervise the usage of sensitive data during the entire life cycle: confidential data and metadata, IoHT’s secrets, and identity (e.g., via user consent).Security: Sensitive data should be securely managed, communicated, stored, processed, transferred, and deleted.Anonymity: Untraceability of identity attributes and unlinkability.Minimality: Sensitive data should be disclosed to a minimum feasible extent for a minimum time only to entities authorized by the user and to be used only for objectives authorized by the user.

Allow the CP-ABE security model to recognize a private key by its set S of defining attributes. To decrypt a message, secret keys must fulfill a policy (access tree) determined by the trusted party expected to encrypt the message.

#### 4.4.1. Security Framework

This subsection discusses the security framework based on the CP-ABE scheme. The elliptic curve EC over a finite field, $F_{q}$, is described by the non-singular curve equation $Y^{2}$ (mod q) = $\left(X^{3}+AX+B\right)$ mod q. In this instance, *A*, *B*, *X*, and *Y* is ∈ $F_{q}$, and the asymmetrical is $4A^{3}+27B^{2}\neq$ 0. Any point Q = $\left(x_{Q},y_{Q}\right)$ on EC has a negativity, which is represented by the equation −Q = $\left(x_{Q},-y_{Q}\right)$ (achieved using the formula $Q+(x_{Q},-y_{Q})=O$). Over E, a cyclic additive group $GE_{c}$ is defined by a *g* generator of order *q*, and $GE_{c}$ then forms an abelian group [18,33,40].

##### Definition of EC-Based-DLP (EC-DLP)

A polynomial-time algorithm cannot encounter the $l\in Z_{q}^{*}$ (selected arbitrarily) such that p = *l*.G. Let two values p, g be $GE_{c}$, where g is the generator of EC group $GE_{c}$. Since *q* is a large prime number, $Z_{q}^{*}$ defines the finite field of integers 1, 2, …, *q* − 1.

##### Definition of EC-Based Comp-DHA (EC-CDHA)

A polynomial-time algorithm encounters computational difficulties in computing s, t, and g, given instance (g, s.g, t.g) $GE_{c}$, where $\left(s,t,\in Z_{q}^{*}\right)$ for a given generator g of $GE_{c}$ (selected randomly).

##### Definition of EC-Based Factorization Problem (EC-FP)

In a polynomial-time algorithm, it is computationally challenging to figure out the values of s.g. and t.g. such that Q = s.g. + t.g., where $s,t\in Z_{q}^{*}$, for a given p and g (selected randomly).

#### 4.4.2. Attacker Model

In this part, we describe various attack scenarios, consider them when developing our attack model, and assess the cloud-based application’s resistance to each one [41,42].

**Assumptions and objectives:** Protecting user data from unwanted access by any third party is the primary security goal. As indicated, the user’s data is kept on a cloud platform. Our system model similarly contains the following trusted, malicious and semi-trusted components.

**Trusted components:** We consider CBs to be reliable and secure. They are employed for the analysis and storage of user data. Before being saved in the CB, the data will be encrypted. The other data center is another reliable element of our system. Using a key security method, CB application policies and data transfers from CB to IoT and blockchain-based applications with authorized users are encrypted [26].

**Semi-trusted component:** The workers in the data center are probably sincere and curious. They adhere to the protocol when registering users and providing them with keys for specific tasks. They want to learn more about using data to alter access rules.

**Malicious components:** In our attack model, we take into account an “external attacker” who is not a member of the system and lacks a valid ID and keys. Additionally, we presume to have an “internal attacker” who is a legitimate system user who has been taken advantage of by an enemy [43].

In our attack model, we assume two attack strategies: (1) an attack on the information’s confidentiality and (2) an attack on its accessibility to the service.

#### 4.4.3. Security Model

Our scheme provides the security model based on the CP-ABE technique to demonstrate the security against a chosen-ciphertext attack [44,45] (Table 3). The relationship between adversary Adv and challenger Chal is explained as in [41].

*Initialization:* adversary Adv outputs the challenge as a piece of access information and transmits it to challenger Chal.

*Setup:* The initialization process is conducted by challenger Chal to provide the system’s required public parameters as the secret key and public pair for the distinct attribute. The opponent Adv receives the shared keys from the challenger.

*Phase 1.* Opponent Adv can foremost create difficulties for the hidden keys of the attributes with the constraint that no bunch of keys can decode the challenge’s coded message. The challenger logs the features in the attribute list that compare to the information provided by the adversary.

*Challenge phase.* At this stage, the adversary Adv generates messages for the challenge as (m1, m2), where m1 and m2 are two equal messages. Then, the challenger chooses a coin c∈(0,1) and transmits the encrypted data to adversary Adv.

*Phase 2.* The same restrictions applied to Phase 1’s secret key queries and decryption questions apply to Phase 2’s opponent, Adv.

*Assume.* The adversary may output a think $c_{0}$ of *c*.

In this game, the adversary’s benefit is described as $|Pr\left[c_{0}=c\right]-1/2|$.


**The CP-ABE scheme structure presents four algorithms: Setup, Encryption, Key Generation, and Decryption for cloud-based applications.**


**SetUp:**$\left\{\lambda \to param,M_{sk}\right\}$ The algorithm produces the output using public parameters *parm* and master private keys $M_{sk}$ using an implicit λ parameter (the ECC domain parameter) and the adaptable set ⋃.

**Encryption:**$\left\{\left(param,msg,\varphi \right)\to C_{ipher}\right\}$: The algorithm encrypts the message *msg* under φ using *param*, *msg*, and φ as inputs. Finally, the algorithm computes the ciphertext $C_{pher}$, and only the user whose attribute array achieves the φ can encrypt the information *msg*.

**Key generation:**$\left\{\left(M_{sk},\upsilon \right)\to P_{k}\right\}$: the algorithm holds the attribute set υ of every user and represented by the key and input master key $M_{sk}$, and generates the secret key $P_{k}$ for the associated user.

**Decryption:**$\left\{\left(param,C_{ipher},P_{k}\right)\to msg\right\}$: The algorithm executes the message *msg* effectively if φ(υ)=1 using *param*, $C_{ipher}$ and $P_{k}$ as input.

## 5. Simulation Results and Analysis

The performance of the proposed approach is examined in this section, which is heavily reliant on the blockchain network, SCs, and smart medical equipment. The effectiveness of the new solutions to problems is verified and evaluated compared to a few of the existing works. Transaction and execution costs are the distinct types of transaction fees on the Ethereum platform in terms of gas. The blockchain network can remove trust issues by reducing transaction costs due to its decentralized network. Transaction costs include the price of sending data, performing procedures, and maintaining contracts. Execution cost is the cost of computing power needed for calculations that store local and global variables. The transaction costs of all smart contacts are shown in Figure 4.

Client- and professional-registration transaction costs are around 198,441 and 198,943 in gas. Client SC monitoring uses less gas than other contracts since fewer inputs are supplied to the contract for monitoring than are provided in the registration fields. The internal storage is more significant the more gas is used during registrations, since the cost increases with the amount of data transferred to the contract. While the total gas utilized and transaction costs for the client-authorization contract are 45,962, they are also lower than the costs associated with registering the patient and the physician, at 44,825, for healthcare management.

In particular, from the perspective of the real-time intelligent healthcare-monitoring system, the response time in the proposed system between submitting an access request and receiving an access response needs to be assessed. Using our testbed, we evaluate the response-time latency of access requests made to our suggested system. The access-request response latency of several existing approaches is compared in Figure 5, showing that as the number of patients rises, the response-time latency slowly increases. The access-request response time for some patient records is limited, according to the performance study of the proposed system, which is shorter than other work already performed.

The access-request response latency of the proposed approach is compared to that of various other existing technologies in Figure 6. It proves that as the number of patients increases, the latency for response time increases gradually. In terms of access-response time, our system performs better than existing schemes.

The deployment and execution times of the suggested model are plotted against the number of patient records in Figure 6. Ten patient-record groupings with 10 to 100 records each exists. For each contract deployed in the first phase, we measure the latency of the intelligent contract deployment. The average time it takes to execute the input functions made available to the contracts, as shown in Figure 6, is the latency of intelligent contract execution. The findings show that as the number of patient health records rises, so does the time required to deploy and execute smart contracts. For instance, the deployment and execution times for 10 patient records are 1.88 and 1.61 s, respectively; for 100 patient records, the times rise to 6.55 and 5.75 s, respectively.

It is essential to consider the CPU and memory resources being used, as a reliable security method should use very little of either of these resources. Measurement statements are added to the running algorithm to achieve this. It is found that the KP-ABE algorithm uses less CPU resources to complete the algorithm’s four stages when compared to CP-ABE and the modified blockchain-based algorithm in terms of CPU and memory usage, and the time complexity of the algorithm is O (1). It should be noted that for setup and key generation, both algorithms use almost the same amount of RAM. The phases of encryption and decryption are slightly different from one another.

## 6. Conclusions

This paper proposes an MLBC-based scheme for a smart-contract-based remote healthcare monitoring system for IoHT. This technique enabled healthcare equipment and the medical approach to secure patient medical records, reduced resources, and improved the IoHT system’s throughput of blockchain. The proposed framework uses the support vector machine to classify, detect and secure, via the CP-ABE scheme, vast amounts of medical’s centers. Additionally, our technology alerts the hospital and doctor of the required medical attention when the patient is in a difficult situation. The Remix framework uses the proposed method to build it on the Ethereum platform. Our suggested approach for secure, distributed, and patient-centric access control is practical in a real-time intelligent healthcare system, as shown by the performance analysis and its efficacy. The system has demonstrated some effective characteristics but suffers from uneven performance and scalability concerns. Decentralized access control is provided by the combination of blockchain and remote patient-monitoring systems. Cloud computing can drastically minimize latency when processing and obtaining patient medical records.

To resolve this issue in the future, more research is, therefore, required. Additionally, future studies will create a methodology for the proposed work’s impetus approach for patient medical data and system architecture for assessing the security of intelligent healthcare equipment using the proper cryptographic methods. To increase the viability and liveliness of the proposed system, an impulsion mechanism for EMR owners and requesters must be established.

## Figures and Tables

**Figure 1 sensors-22-09074-f001:**
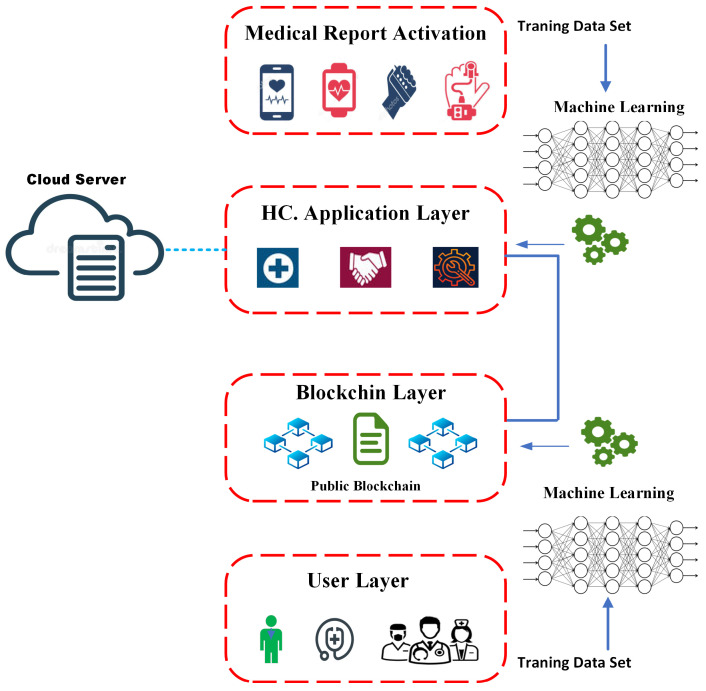
Cloud-Based Machine Learning and Blockchain.

**Figure 2 sensors-22-09074-f002:**
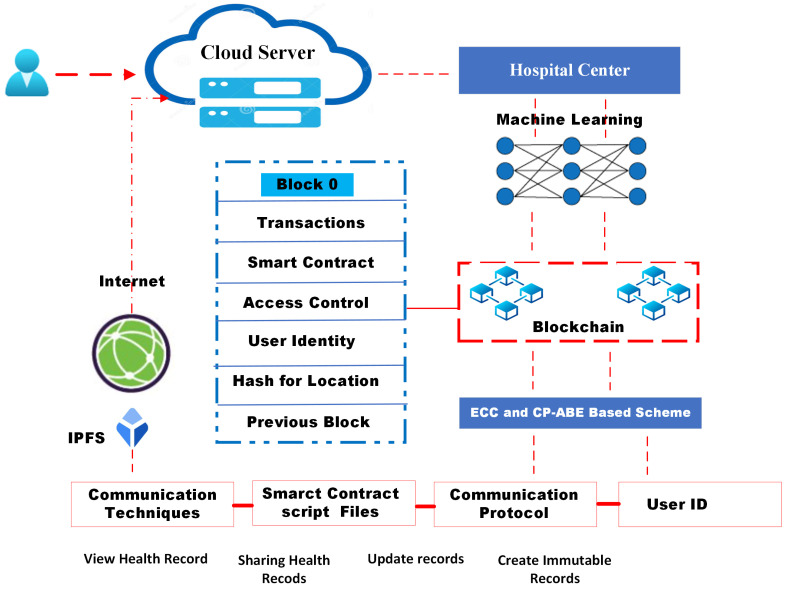
Secure machine-learning-based blockchain model for IoHT.

**Figure 3 sensors-22-09074-f003:**
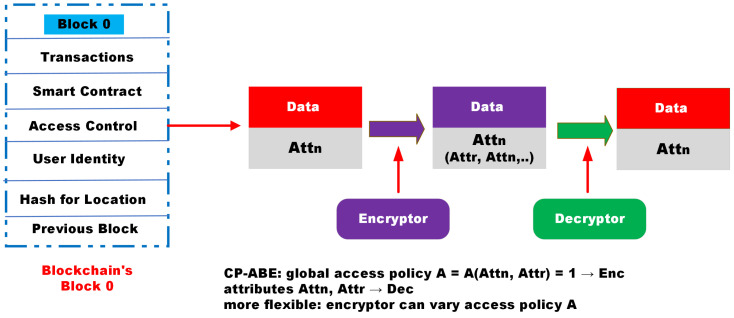
Attribute-based encryption/decryption techniques in blockchain.

**Figure 4 sensors-22-09074-f004:**
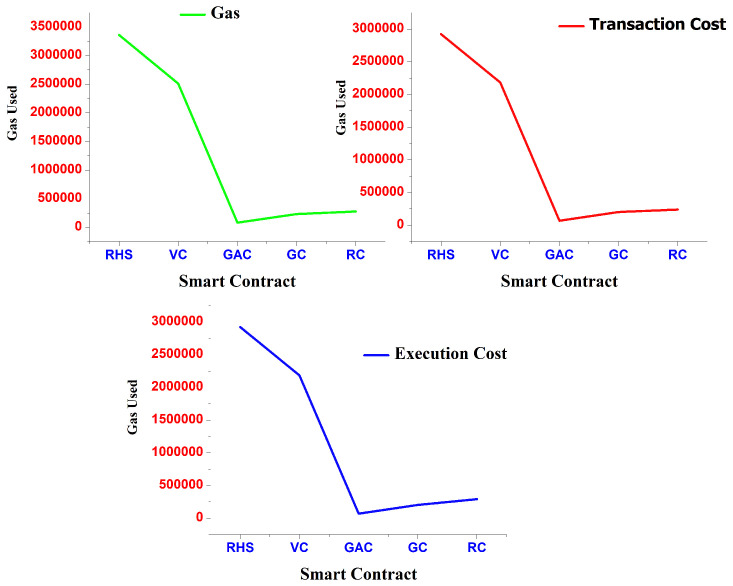
Smart contract using GAS, transaction and execution cost.

**Figure 5 sensors-22-09074-f005:**
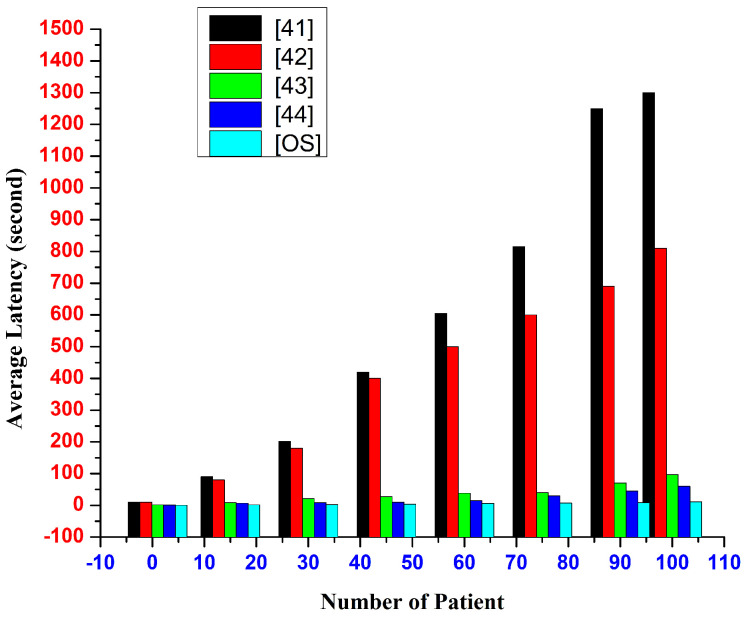
Number of patient requests and response latency in hospital center [41,42,43,44].

**Figure 6 sensors-22-09074-f006:**
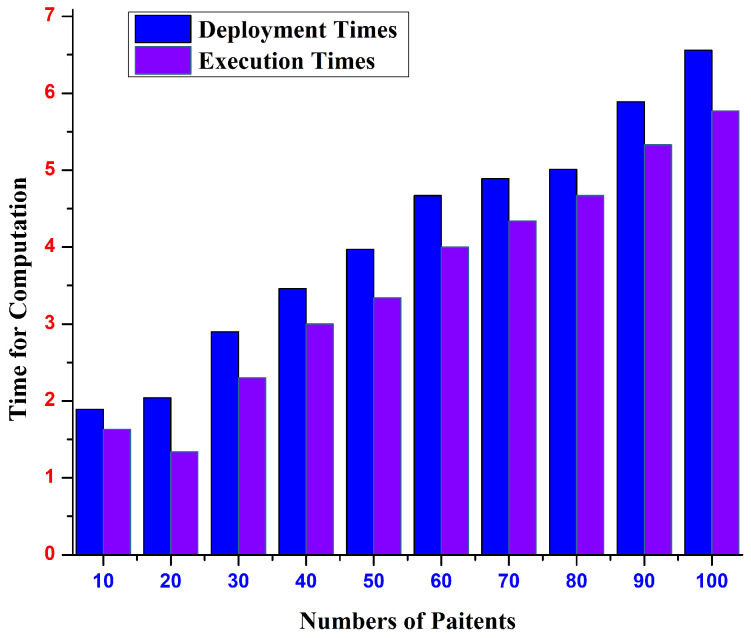
ML-based blockchain execution and deployment of smart contracts.

**Table 1 sensors-22-09074-t001:** Comparison of various schemes’ security, classification and detection approach.

Scheme	Byte Length	Accuracy	Approach	Data Used	Goal	Classes	Platform	Data Size
[26]	2KB	94 %	DL	Bytes	DET	2	IoMT	36,236
[26]	2KB	95 %	DL	Bytes	CLS	5	IoMT	36,236
[26]	2KB	94 %	DL	Bytes	CLS	10	IoMT	36,236
[27]	-	98 %	CNN	CFG	DET	2	IoT	11,200
[28]	-	97 %	CNN	CFG	CLS	2	IoT	6000
[28]	-	98%	CNN	CFG	CLS	3	IoT	6000
[29]	-	96%	FS	Opcodes	DET	2	IoT	1207
[30]	-	95%	XGBoost	Opcodes	DET	2	IoT	4169
[30]	128 to 1024	98%	SVM	Bytes	DET	2	IoT	222K
[31]	128 to 1024	96%	SVM	Bytes	CLS	8	IoT	222K
[31]	-	98%	SVM	CFG	DET	2	IoT	5476
[31]	-	97%	SVM	CFG	DET	2	IoT	6560

DL: deep learning, FS: fuzzy system, DET: detection, CLS: classification.

**Table 2 sensors-22-09074-t002:** Comparison of various schemes using blockchain-based security with different platforms.

Schemes	Year	Problem Identification	Technique Used	Platform Used
[7]	2020	The present edge computing technologies are unable to give ultra-low latency for real-time applications.	Proposed three-tier architecture with home edge computing (HEC)	IoT
[9]	2018	Existing security systems are unable to successfully address the unique security issues arising in smart homes.	Examined the probability of blockchain as an answer to the security of smart homes. Proposed a simple but secured framework for smart homes using consortium blockchain.	Blockchain
[10]	2020	Use of a single server is unable to manage a number of devices with expected high-level efficiency and also the system suffers from malicious devices.	Hybrid blockchain-based physical-authentication scheme for smart homes suggested	Blockchain
[11]	2020	The privacy, security and the reliability of IoT applications are key challenges.	Blockchain provides a decentralized and transparent database platform for security at a higher level.	Blockchain
[12]	2021	Large-scale-use IoT system which employs data-sharing system based on a centralized cloud which creates trust issues.	Uses blockchain-based marketplace to transfer data and proxy re-encryption system to provide appropriate information transfer	IoT
[13]	2021	IoT devices have insufficient computing complexity for supporting robustness and encryption algorithms.	Suggest embedding a blockchain-protected interface into an IoT-based appliance protection gateway. It also provides authentication of data and decentralization in the system.	Blockchain
[14]	2021	Conventional smart homes have challenges in security domains such as data readability.	To ensure proper security in IoT devices, Ethereum blockchain can be used and also consensus play a vital role.	Blockchain
[17]	2018	IoT smart homes have appliances that capture environmental information, and so have chances of malicious attacks and privacy issues.	According to a thorough analysis of blockchain technology regarding IoT-based smart homes, cloud computing reinforced via blockchain with the disseminated system are secure, andpersonal qualities can offer a solid solution.	Blockchain
[15,16]	2019	IoT smart homes have faced problems regarding data storage as the data stored are very confidential.	Provides a three-level architecture with the best security possible for managing the data storage of smart homes through cloud service providers based on blockchain.	IoT
[32]	2022	Multiple attribute authorities are used in the technique to handle attributes and key generation, which helps lessen the workload associated with using a single authority in conventional CP-ABE systems.	The key-escrow issue is resolved by the plan as well. Here, outsourcing the decryption procedure to a data-user assistant reduces the computing burden on the end users.The security analysis demonstrates that the suggested system is resistant to forgery, collusion, andman-in-the-middle attacks.	IoT
[33]	2022	The research can be utilized to calculate road-condition early notification systems based on blockchain for automobile self-organizing networks based on the actual metropolitan traffic circumstances, theroadside unit, andthe cloud-server deployment plan.	To further enhance intelligent mobility systems in smart cities, blockchain is being investigated for use in vehicle management based on Digital Twins in Vehicular Adhoc Networks (VANETs).	VANETs

**Table 3 sensors-22-09074-t003:** Comparison of various security schemes and proposed scheme with different security services.

Scheme	Data Security	Integrity	Authentication	Privacy	Anonymity	Inf. Leakage Prev.
[46]	🗸	🞨	🗸	🗸	🞨	🗸
[47]	🗸	🗸	🗸	🞨	🗸	🗸
[48]	🗸	🗸	🗸	🗸	🗸	🗸
[49]	🗸	🞨	🗸	🗸	🗸	🗸
[32]	🞨	🗸	🗸	🗸	🗸	🞨
[45]	🗸	🞨	🗸	🗸	🞨	🗸
Our scheme	🗸	🗸	🗸	🗸	🗸	🗸

## Data Availability

Data will be made available on request.

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
