# Peer review of "A Machine-Learning–Blockchain-Based Authentication Using Smart Contracts for an IoHT System"

_sensors, 2022, doi:10.3390/s22239074_

Round 1
Reviewer 1 Report
The authors have proposed a machine learning-blockchain-based authentication using a smart
contract for the IoHT system. The problem solved in the article is significant and relevant to the
venue. The scientific novelty and contributions of this paper are good enough for publication.
However, I have a few suggestions worth including to represent the article better.
Positive aspects of the paper:
1. This paper is well-motivated and easy to follow.
2. Experiments are well-designed, and the proposed algorithm works well compared to baseline
algorithms.
Suggestions/comments:
1. The caption texts in some figures are hard to read and require an increased font size. The
reviewer suggests using a different table for symbols and abbreviations.
2. What issues are missing in solving previous approaches to the scheme? What are the
drawbacks or weaknesses of the previous approaches?
3. The manuscript is valuable to the literature and adds a few more related works.
4. The main contributions of this paper should be further summarized and demonstrated. This
reviewer suggests that the authors explain what is new about their approach and why it should be
used instead of the ones already out there.
5. Cite some of the recently published articles relevant to the topic published such as the
following “On the design of blockchain-based ECDSA with fault-tolerant batch verification protocol for blockchain-enabled IoMT”
6. The paper needs proofreading to eliminate any typos or grammatical errors.
7. Compare the results obtained with recent state of the art.
8. A detailed analysis on the results obtained has to be presented.
9. How did the authors choose the hyper-parameters of the proposed ML algorithm? Did they used any parameter tuning method or is it random?
10. The authors have to extend the results section by presenting an analysis on the time complexity of the proposed approach.
11. What are the changes that have to be made to make the proposed approach to adapt to real-time health monitoring?
Author Response
Journal: Sensors
Manuscript ID: sensors-2042273
Title: A Machine Learning- Blockchain Based Authentication Using Smart Contract for IoHT System
Review Report Form # 1
Open Review ( ) I would not like to sign my review report
(x) I would like to sign my review report
English language and style
( ) English very difficult to understand/incomprehensible
( ) Extensive editing of English language and style required
( ) Moderate English changes required
(x) English language and style are fine/minor spell check required
( ) I don't feel qualified to judge about the English language and style
Author Response: Thank you for your comments and for supporting us in enhancing the manuscript. We have used professional English language grammar and spelling services to ensure the research article does not have any language issues.
|
|
|
Can be improved |
Must be improved |
Not applicable |
|
Does the introduction provide sufficient background and include all relevant references? |
( ) |
(x) |
( ) |
( ) |
|
Are all the cited references relevant to the research? |
( ) |
(x) |
( ) |
( ) |
|
Is the research design appropriate? |
( ) |
(x) |
( ) |
( ) |
|
Are the methods adequately described? |
( ) |
(x) |
( ) |
( ) |
|
Are the results clearly presented? |
( ) |
(x) |
( ) |
( ) |
|
Are the conclusions supported by the results? |
( ) |
(x) |
( ) |
( ) |
Comments and Suggestions for Authors
The authors have proposed a machine learning-blockchain-based authentication using a smart contract for the IoHT system. The problem solved in the article is significant and relevant to the venue. The scientific novelty and contributions of this paper are good enough for publication.
However, I have a few suggestions worth including to represent the article better.
Author Response:
Thank you for your positive and encouraging comments and your support to help enhance our manuscript.
Positive aspects of the paper:
- This paper is well-motivated and easy to follow.
- Experiments are well-designed, and the proposed algorithm works well compared to baseline algorithms.
Author Response:
Thank you for your encouraging and positive comments and for helping us improve our manuscript.
Suggestions/comments:
- The caption texts in some figures are hard to read and require an increased font size. The reviewer suggests using a different table for symbols and abbreviations.
Author Response:
Your comments and suggestions for improving the manuscript are much appreciated. As per the reviewer’s comments, we have increased the font size of caption text used for figures in the manuscript to improve the readability of the paper. The change is highlighted in the paper. Also, as per the suggestion of the reviewer, we have used a separate table for symbols and abbreviations in the revised manuscript.
- What issues are missing in solving previous approaches to the scheme? What are the
drawbacks or weaknesses of the previous approaches?
Author Response:
Your comments and suggestions for improving the manuscript are much appreciated. The various previous approaches do not support some security services like integrity, authentication, data security, privacy, and anonymity. Our proposed scheme supports all the security services that are mentioned in Table 3.
- The manuscript is valuable to the literature and adds a few more related works.
Author Response:
Thank you for your comments and for supporting us in enhancing the manuscript. As per the suggestion of the reviewer, we have added some recent papers in related work. The changes are highlighted in the revised manuscript.
- The main contributions of this paper should be further summarized and demonstrated. This reviewer suggests that the authors explain what is new about their approach and why it should be used instead of the ones already out there.
Author Response:
Thank you for your comments and for supporting us in enhancing the manuscript. As per the suggestion of the reviewer, the main contribution of this work is updated to highlight the author’s real contribution to this work. In our proposed approach machine learning and blockchain is integrated to make more effective and efficient data storage and processing. The change is highlighted in the revised manuscript.
- Cite some of the recently published articles relevant to the topic published such as the
following “On the design of blockchain-based ECDSA with fault-tolerant batch verification protocol for blockchain-enabled IoMT”.
Author Response:
We appreciate your feedback and assistance in revising the manuscript. As per the reviewer’s suggestion, the related works are studied, and the above-mentioned paper is cited in the revised manuscript. The new paragraphs for related work are added to the revised paper. The change is highlighted in the revised manuscript.
- The paper needs proofreading to eliminate any typos or grammatical errors.
Author Response:
Your comments and suggestions for improving the manuscript are much appreciated. As per the reviewer’s comments, we have proofread all the content of paper and eliminated all typo and grammatical errors.
- Compare the results obtained with recent state of the art.
Author Response:
Your comments and suggestions for improving the manuscript are much appreciated. As per the reviewer’s comments, we have compared the result of our proposed scheme with those of other schemes. Our proposed scheme is better than other existing schemes, as shown in table 3.
- A detailed analysis on the results obtained has to be presented.
Author Response:
Your comments and suggestions for improving the manuscript are much appreciated. A detailed analysis of the result is already presented in figures 4, 5, and 6 in Section 5.
- How did the authors choose the hyper-parameters of the proposed ML algorithm? Did they used any parameter tuning method or is it random?
Author Response:
Your comments and suggestions for improving the manuscript are much appreciated. In our proposed approach, machine learning is used for classification of data. We have not used any parameter tunning in this approach. We adopted it randomly.
- The authors have to extend the results section by presenting an analysis on the time complexity of the proposed approach.
Author Response:
Thank you for your comments and for supporting us in enhancing the manuscript. We have added a discussion about time complexity in result section. The changes are highlighted in the revised manuscript.
- What are the changes that have to be made to make the proposed approach to adapt to real-time health monitoring?
Author Response:
Your comments and suggestions for improving the manuscript are much appreciated. We have used machine learning – blockchain based technique in our proposed approach. Blockchain allows for the electronic distribution of data storage in a cloud-based system, enabling anytime access to the data through a cloud server or ledger system. Each block used in this method can carry a specific quantity of data. Our proposed approach is used in real-time health monitoring because it allows access to data at any time.
Reviewer 2 Report
This paper presents a machine learning- Blockchain based authentication using smart contract for IoHT System. Authors propose a BCML-based multidimensional approach architecture for IoHT, malware detection, classification, and security scheme for healthcare systems. In addition, The machine Learning layer integrates into the proposed security model, transforming the data into the IoHT application system.
This is an interesting work, However, I have a few comments:
1. Contributions are not mentioned in the abstract.
2. In the introduction: The comparison with previous works must be more precise in order to highlight the real contribution of this work.
3. English must be revised. The manuscript should be formatted better and some spelling and grammar should be checked carefully.
4. Figure 2 can be removed.
5. Section 3.3. A Comparative Study and Discussion : This part is very abstracted, please add more discussion to enrich it.
6. Section 4.4.3. Security Model : how did you generate the table 3 ?
Concluding, the paper has potential to be appreciated by the readers and the above comments are formulated such that to enhance its impact.
Author Response
Journal: Sensors
Manuscript ID: sensors-2042273
Title: A Machine Learning- Blockchain Based Authentication Using Smart Contract for IoHT System
Review Report Form # 2
Open Review (x) I would not like to sign my review report
( ) I would like to sign my review report
English language and style
( ) English very difficult to understand/incomprehensible
( ) Extensive editing of English language and style required
(x) Moderate English changes required
( ) English language and style are fine/minor spell check required
( ) I don't feel qualified to judge about the English language and style.
Author Response:
Thank you for your comments and for supporting us in enhancing the manuscript. We have used professional English language grammar and spelling services to ensure the research article does not have any language issues.
|
|
|
Can be improved |
Must be improved |
Not applicable |
|
Does the introduction provide sufficient background and include all relevant references? |
( ) |
( ) |
(x) |
( ) |
|
Are all the cited references relevant to the research? |
( ) |
(x) |
( ) |
( ) |
|
Is the research design appropriate? |
( ) |
(x) |
( ) |
( ) |
|
Are the methods adequately described? |
( ) |
( ) |
(x) |
( ) |
|
Are the results clearly presented? |
( ) |
( ) |
(x) |
( ) |
|
Are the conclusions supported by the results? |
( ) |
( ) |
(x) |
( ) |
Comments and Suggestions for Authors
This paper presents a machine learning- Blockchain based authentication using smart contract for IoHT System. Authors propose a BCML-based multidimensional approach architecture for IoHT, malware detection, classification, and security scheme for healthcare systems. In addition, The machine Learning layer integrates into the proposed security model, transforming the data into the IoHT application system.
This is an interesting work; However, I have a few comments:
Author Response:
Thank you for your positive and encouraging comments and your support to help enhance our manuscript.
- Contributions are not mentioned in the abstract.
Author Response:
Thank you for your feedback and for assisting us in revising the manuscript. As per the suggestion of the reviewer, the abstract is updated to include the contributions of the authors in the revised manuscript. The changes are highlighted in the paper.
- In the introduction: The comparison with previous works must be more precise in order to highlight the real contribution of this work.
Author Response:
Thank you for your comment and for supporting us in enhancing the manuscript. As per the suggestion of the reviewer, in the introduction section, ‘our contribution’ is updated to highlight the author’s real contribution to this work.
- English must be revised. The manuscript should be formatted better and some spelling and grammar should be checked carefully.
Author Response:
Thank you for your comment and for supporting us in enhancing the manuscript. As per the reviewer comments, English is revised, and spelling and grammar are checked and corrected in the revised manuscript.
- Figure 2 can be removed.
Author Response:
Thank you for your comments and for supporting us in enhancing the manuscript. As per the reviewer suggestion, figure 2 is removed from the manuscript and the rest of the figure numbers are updated.
- Section 3.3. A Comparative Study and Discussion: This part is very abstracted, please add more discussion to enrich it.
Author Response:
Your comments and suggestions for improving the manuscript are much appreciated. As per reviewer suggestion, we have added more discussion in section 3.3. We have added a paragraph to describe table 2. Also, some recent papers are added in table 2 to compare various security schemes. The changes are highlighted in the revised manuscript.
- Section 4.4.3. Security Model: how did you generate the table 3?
Author Response:
Thank you for your comments and for supporting us in enhancing the manuscript. Table 3 describes the comparison of various security schemes and the proposed scheme with different security services. To generate table 3, we have analysed many papers and found that security schemes in some papers support and some do not support security services like data security, integrity, authentication, privacy, anonymity and information leakage prevention. The security services that scheme supports are denoted by (√), while those that it does not support are denoted by (x). Our proposed scheme is better than other existing schemes, as shown in table 3.
Round 2
Reviewer 1 Report
All the comments are addressed.
Reviewer 2 Report
The authors reacted properly to my pointed issues.